

# An artificial neural network for automated behavioral state classification in rats

Jacob G. Ellen[1] and Michael B. Dash[1,2]

[1] Neuroscience Program, Middlebury College, Middlebury, VT, United States
[2] Psychology Department, Middlebury College, Middlebury, VT, United States

## ABSTRACT

Accurate behavioral state classification is critical for many research applications. Researchers typically rely upon manual identification of behavioral state through visual inspection of electrophysiological signals, but this approach is time intensive and subject to low inter-rater reliability. To overcome these limitations, a diverse set of algorithmic approaches have been put forth to automate the classification process. Recently, novel machine learning approaches have been detailed that produce rapid and highly accurate classifications. These approaches however, are often computationally expensive, require significant expertise to implement, and/or require proprietary software that limits broader adoption. Here we detail a novel artificial neural network that uses electrophysiological features to automatically classify behavioral state in rats with high accuracy, sensitivity, and specificity. Common parameters of interest to sleep scientists, including state-dependent power spectra and homeostatic non-REM slow wave activity, did not significantly differ when using this automated classifier as compared to manual scoring. Flexible options enable researchers to further increase classification accuracy through manual rescoring of a small subset of time intervals with low model prediction certainty or further decrease researcher time by generalizing trained networks across multiple recording days. The algorithm is fully open-source and coded within a popular, and freely available, software platform to increase access to this research tool and provide additional flexibility for future researchers. In sum, we have developed a readily implementable, efficient, and effective approach for automated behavioral state classification in rats.

## INTRODUCTION

State-dependent differences in neuronal activity contribute to widespread changes in information processing and conscious awareness and ultimately appear to subserve many important functions for maintaining brain health (*Haglund, Pavan & Nedergaard, 2020*; *Nir et al., 2013*; *Quilichini & Bernard, 2012*). Characterizing and identifying these state-dependent activity patterns is therefore a critical task for addressing a diverse set of research questions. Manual classification of behavioral state using visual scoring of the electroencephalogram (EEG), local field potential (LFP), and/or electromyogram (EMG) is often performed, yet has significant limitations as compared to algorithmic classifiers.

Corresponding author
Michael B. Dash,
mdash@middlebury.edu

Specifically, manual classification is time intensive and scales proportionally with the size of a data set, ultimately relies upon subjective determinations that limit inter-rater reliability, and requires extensively trained researchers to achieve high classification accuracies (*Gao, Turek & Vitaterna, 2016*; *Gross et al., 2009*; *Miladinović et al., 2019*; *Rytkönen, Zitting & Porkka-Heiskanen, 2011*; *Stephenson et al., 2009*).

Many promising automatic behavioral state classifiers have been developed in recent years. Although each of these classifiers typically relies upon characteristics derived from similar electrophysiological signals (*i.e.* EEG, LFP and EMG), they differ as a function of how those characteristics are selected and combined to predict behavioral state. Semi-automated approaches, including user defined logic rules/thresholds (*Gross et al., 2009*) and principal component analyses (*Gilmour et al., 2010*), accurately predict behavioral state using researcher-identified electrophysiological features. Supervised learning algorithms similarly classify behavioral state based on handcrafted features and expert classifications, but are able to automatically "learn" rules for combining those features to accurately predict behavioral state. Successful supervised classifiers have been implemented through support vector machines (*Crisler et al., 2008*), naïve Bayes classifiers (*Rempe, Clegern & Wisor, 2015*; *Rytkönen, Zitting & Porkka-Heiskanen, 2011*), decision trees and linear discriminant analysis (*Brankack et al., 2010*) and an ensemble method that uses all of the above (*Gao, Turek & Vitaterna, 2016*). More recent supervised approaches have utilized deep learning algorithms which use large amounts of training data and computational power to produces predictions without the need for initial feature selection (*Barger et al., 2019*; *Exarchos et al., 2020*; *Miladinović et al., 2019*; *Yamabe et al., 2019*). Each of these approaches differ from unsupervised learning algorithms which find undetected patterns in the dataset without the need for initial feature identification or manual scoring from experts (*Sunagawa et al., 2013*; *Yaghouby, O'Hara & Sunderam, 2016*).

Despite the excellent performance of the automated behavioral state classifiers highlighted above, many researchers are still dependent upon time-intensive manual classification. Significant functional barriers exist for implementing these automated approaches that typically require extensive computational resources, are constructed within proprietary software, and/or still require significant expert specification of features. To overcome these potential limitations, we developed an artificial neural network (ANN) algorithm for behavioral state classification within the popular, and open-source, R software platform (*R Core Team, 2020*). This supervised classifier is comprised of four consecutive feed-forward, fully connected layers of artificial neurons. Computational weights of the network are repeatedly adjusted during training in order to help the network 'learn' to classify behavioral state using electrophysiological features of individual 4 s time intervals. This novel ANN: (1) is easy to implement, (2) has high classification accuracy, sensitivity, and specificity that are comparable to leading algorithms, (3) has low computational complexity thereby enabling fast algorithm performance with minimal computational resources, and (4) affords flexibility for individual researchers by using freely-available code that is readily customizable in a popular open-source environment. Consequently, this tool serves as a highly accessible and effective automated behavioral state classifier.

## MATERIALS AND METHODS

### Animal care and use

All sample data were obtained from previous experiments conducted at Middlebury College with all methods performed in accordance with the National Institutes of Health Guide for the Care and Use of Laboratory Animals and approved by Middlebury College's Institution Animal Care and Use Committee (approved research protocol #316-17). Male, Sprague-Dawley rats (3–4 months old, Charles River, Wilmington, MA, USA) were obtained and pair-housed upon arrival in clear, plastic rodent caging (16 in. × 7.5 in. × 8 in.; Teklad TEK-Fresh bedding, Envigo, Indianapolis, IN, USA). Food (Teklad 2020X) and water were provided *ad libitum* along with in-cage enrichment objects (wooden blocks, pvc tubing, paper towels). After an initial acclimation period (>1 week from initial arrival to the vivarium), each rat underwent stereotactic surgery. Here, rats were anesthetized *via* isoflurane (3.5% induction, 2–3% maintenance) and given pre-operative analgesic (Meloxicam; 2 mg/kg; MWI, Boise, ID, USA) and antibiotic (Penicillin, 100,000 units/kg) treatments. During surgery, electrodes were implanted for EEG/LFP and nuchal EMG recordings (see sample data set below for additional recording characteristics) and were affixed to the skull using dental acrylic (Lang Dental; Wheeling, IL, USA). A postoperative analgesic (Meloxicam, 2 mg/kg) was administered between 12–24 h after surgery completion. For the duration of the experiment, rats were then single-housed to minimize potential damage to implanted electrodes and the headstage preamplifier (100× amplification, Pinnacle Technologies, Lawrence, KS, USA). Throughout the experiment, each rat was monitored daily for overt signs of pain/distress including immobility, poor grooming, weight loss, porphyrin staining, postural abnormalities, lack of food/water consumption, and signs of infection around the surgical site. Observation of these signs leads to direct consultations with the attending veterinarian and vivarium staff to determine the appropriate course of action (*e.g.* treatment or euthanasia). In the present study, no rats were euthanized prior to the conclusion of the experiment. In accordance with the 2013 AVMA Guidelines of Euthanasia, all rats were euthanized *via* $CO_2$ exposure at the end of the experiment.

### Sample data set

Uninterrupted baseline recording days ($N = 50$ from 11 total rats) were selected from male, Sprague-Dawley rats (3–4 months old). Care was taken to generate a sample of recording days and rats that collectively comprise common recording characteristics across rodent sleep research. Specifically, within each rat, two electrophysiological signals to measure brain activity and one nuchal electromyogram (EMG) were recorded (sampling rate = 250 Hz; 8401 DACS; Pinnacle Technologies, Lawrence, KS, USA). Rats used in our sample data set had a wide range of recording locations (*e.g.* prefrontal cortex, motor cortex, parietal cortex) and diverse recording modalities (2 EEGs, 1 EEG/1LFP, or 2 LFPs). Likewise, these data differed in overall signal quality (*e.g.* 4.83 ± 0.56% mean daily artifact prevalence; range: 0.15% to 13.48%). Consequently, this data set represents typical electrophysiological recordings of rodent sleep. Of note, two rats were ultimately excluded

from analyses because of very poor EMG signal quality resulting in a final sample data set of 40 baseline days recorded across nine rats. Details for how EMG signal quality was assessed are presented below as part of our description of our artificial neural network for automated sleep classification.

Trained undergraduate sleep researchers visually scored behavioral state offline in 4 s time intervals for each baseline recording day (following conventional sleep research terminology, 4 s time intervals will be referred to as epochs throughout the remainder of the manuscript). Epochs were classified as waking when containing low-voltage, high-frequency EEG/LFP activity and elevated EMG. High-voltage, low-frequency EEG/LFP activity and an absence of EMG activity was characteristic of NREM sleep, while REM sleep epochs were classified when low-voltage, high-frequency EEG/LFP activity and an absence of EMG activity were observed.

## Artificial neural network

To develop an effective research tool with minimal entry barriers for use, we sought to design a freely available, open-source implementation to automatically classify behavioral state using electrophysiological features. With this goal in mind, we chose to implement our algorithm using R (*R Core Team, 2020*), a freely-available statistical computing platform that has been widely adopted across scientific disciplines (*e.g.* 58% of ecological papers published in 2017 used R as their primary analytic tool *Lai et al., 2019*). Instructions for full implementation of our algorithm and source code can be found at https://github.com/jellen44/AutomaticSleepScoringTool.

Within R, we implemented an artificial neural network (ANN) with four, fully connected sequential layers (256, 128, 32 and 3 nodes) to automatically classify behavioral state in freely behaving rodents. Our ANN was implemented using the Keras package in R with a Tensorflow backend (*Abadi et al., 2016*). In each of the hidden layers, a weighted sum of inputs is computed, followed by an activation function rectified linear unit activation (ReLU) that introduces nonlinearity and regulates neuronal activation. These weights are adjusted during the training process to improve classification performance. The output layer of this network takes input from the last hidden layer and outputs a vector containing three values, each representing the probability that the epoch is one of the three sleep states. We accomplished this with the SoftMax function, which is commonly used in multiclass classification problems because it takes a K-dimensional vector (the final layer in our network) as an input and uses it to estimate a range of probabilities over a given number of classes (*Duan et al., 2003*).

### Feature extraction

Input for the artificial neural network consisted of 13 EEG-based features per EEG, two EMG features, and two features derived from both EEG and EMG activity. To extract EEG features, we calculated power spectra (Welch's method, Hamming window) for each 4 s epoch. From the resultant power spectra, we calculated band-limited power (BLP) within the following frequency bands (delta, 1–4 Hz; theta 4–7 Hz; upper theta 7–9 Hz; alpha 8–12 Hz; beta 13–30 Hz, low gamma 30–50 Hz, medium gamma 50–75 Hz, high

gamma 76–125 Hz). Five additional features were generated as ratios between frequency bands: (1) beta/delta, (2) beta/low gamma, (3) beta/high gamma, (4) theta/delta, and (5) theta/medium gamma. EMG features consisted of the root-mean square (RMS) of raw EMG activity and EMG power. As both EMG and gamma activity (*Brankack et al., 2010*) may serve as useful features for classification of REM sleep, two additional features were generated by summing each EMG feature with medium gamma BLP. Ultimately, all features were z-score normalized prior to input into the ANN model.

### ANN training and testing

ANN models can operate as supervised learning algorithms that rely upon an initial training set to make predictions about remaining test cases. Our ANN model requires a small initial training set (560, 4 s epochs; *i.e.* 2.6% of daily total) to achieve high classification performance (see Fig. S1). To increase the likelihood that training features are representative of the diversity of test case activity, we first manually scored 50 pseudorandom series of 10 epochs each. Here random series were selected within the following criteria: (1) they were derived from time points throughout the 24 h period and (2) periods of NREM sleep, REM sleep, and waking were all included across the 50 series selected. Due to the naturally low prevalence of REM epochs, an additional 60 REM epochs were specifically sought out, manually scored, and incorporated into this training set to ensure sufficient input across all three behavioral states is present for the model. This training set was then algorithmically oversampled (minority classes resampled with replacement) to provide a complete training set with an equal number of epochs across the three behavioral states to be classified in order to avoid highly unbalanced classes due to the low number of REM epochs.

During training, the ANN weights are modified to minimize a categorical cross entropy loss function (a common loss function for discrete classification tasks). To prevent overfitting, we implemented Ridge (L2) regularization (β = 0.01), which encourages small-weight values by penalizing larger weight values through the loss function (*Janocha & Czarnecki, 2017*). Training occurred across 100 complete passes of the training data set (*i.e.* machine learning epochs) with a batch size of 10 samples. The adaptive moment estimation (Adam) optimizer was used to minimize the loss function with a learning rate of 0.001 to balance accuracy with training time.

Stochastic effects during model training can result in identical inputs producing functional differences in the trained ANN. We therefore implemented an ensemble learning approach (*Sagi & Rokach, 2018*), in which outputs from multiple models are combined to improve predictive performance. For each day, the above training procedures were repeated five times and thereby generated five similar, albeit distinct ANNs. Electrophysiological features derived from each 4 s epoch not contained within the training set were used as input features to test the efficacy of the ANNs for classifying behavioral state. Behavioral state was determined as the modal predicted value for each epoch across the five trained ANNs (*i.e.* most frequent prediction). By averaging prediction certainty across the five ANNs, we also produced a final estimate of the likelihood that a given test epoch was reflective of each behavioral state. For epochs where the mean

classification probability across the five ANN models was under 90%, an "uncertain" label was added to that epoch (in addition to the most likely categorical behavioral state output). As detailed more fully in the results section below, this "uncertain" label enables researchers to easily identify a small subset of epochs that (1) are misclassified at a higher rate by the automated ANN and (2) as an option, can be manually rescored to increase classification accuracy while still relying upon automated classification for the vast majority of epochs.

As the above ANNs are trained and tested on an epoch-by-epoch basis (with no between epoch history), these models are unable to directly incorporate sleep/wake history into their behavioral state predictions. To overcome this limitation, once all test epochs have been predicted, we applied a final series of simple heuristic rules to amend the ANN behavioral state predictions to better reflect typical sleep/wake histories. Specifically, we increased REM sleep continuity by rescoring small bouts (<=12 s) of scored waking or NREM sleep located between otherwise continuous REM epochs as REM sleep. As direct transitions from waking to REM sleep are exceptionally rare except under pathological conditions (*Fujiki et al., 2009*; *Mignot et al., 2006*), we additionally rescored any epochs of REM sleep that occurred immediately following waking as waking. Lastly, we identified isolated 4 s epochs whose state scoring was bounded by epochs that (a) differed from the isolated epoch and (b) were otherwise scored identically. Once identified, these isolated epochs were rescored to match the immediately preceding/ensuing epochs.

Through the course of training and testing these ANNs across our sample data set, it became apparent that the predictive performance of these networks for classifying REM sleep was severely impaired in the absence of a well-functioning EMG. Consequently, prior to training our algorithm first calculates the EMG coefficient of variation (EMG standard deviation/mean RMS EMG). If this standardized quality metric is less than 1.67 (an experimentally derived threshold that appears to identify poor REM sleep classification; see Fig. S2), a warning to the user that the automated state scoring algorithm is unlikely to successfully score the REM epochs of that particular file is generated. As indicated above, this procedure resulted in the removal of data from two rats in our original sample set.

## Analyses and statistical approaches

All analyses of model performance were conducted in Mathworks Matlab (Natick, MA, USA) with additional statistical analyses (correlation, repeated-measures ANOVA, *t*-tests) conducted in SPSS (IBM, Armonk, NY, USA). All data are presented as mean ± standard error.

Performance of the automated sleep scoring algorithm was assessed by comparing its predictions of behavioral state for each epoch with corresponding epochs classified manually. Key metrics calculated include overall accuracy (% of epochs that were scored identically by automated and manual approaches), state-dependent sensitivity (TP/TP + FN), and state-dependent specificity (TN/TN + FP); where TP, true positive; FN, false negative; TN, true negative, and FP, false positive. To determine the extent to which training the ANN using data from one baseline day could generalize to (a) other baseline days from the same rat and (b) other baseline days from different rats, we likewise

calculated accuracy, sensitivity, and specificity of the algorithm under these conditions. For within rat generalization, ANNs that were separately trained from each baseline day in the data set were used to predict behavioral state for all other baseline days recorded from that rat. For between rat generalization, ANNs that were separately trained from each baseline day in the data set were used to each predict behavioral state for five baseline days randomly selected from other rats.

Lastly, we calculated the effects of using ANN classification on common sleep parameters. Power spectra were calculated for each 4 s epoch using Welch's method (Hamming window) and averaged across all epochs of the same behavioral state. To account for potential differences in overall signal strength, spectra were normalized to mean broadband power (*i.e.* from 0.5–125 Hz). To calculate slow wave activity (SWA), we calculated band limited power (0.5–4 Hz) for each 4 s epoch of NREM sleep and averaged these values across each hour of the light period. Hourly SWA values were then normalized to the mean SWA across the light period.

## RESULTS

### Electrophysiological characteristics associated with behavioral state are useful features for automated state classification

Behavioral state scoring, be it manual or algorithmic, relies upon the distinct patterns of neuronal activity characteristic of each behavioral state. Waking epochs are typified by low-voltage, high-frequency EEG activity and variable EMG activity. By contrast, as a function of the large slow waves that predominate NREM sleep, EEG activity during NREM epochs is characteristically high-voltage and low-frequency. REM sleep meanwhile, exhibits EEG activity that is similar to that of waking, but with the complete absence of EMG activity due to REM-associated motor atonia. While these electrophysiological differences can typically be discerned visually in the time-domain (see Fig. 1A for example tracings), quantification of state-dependent activity is usually performed through the extraction of key frequency-domain features. As evident in Fig. 1B, EEG band-limited power (BLP) likewise exhibits pronounced state dependency. For example, delta BLP (0.5–4 Hz; *i.e.* low-frequency) is clearly elevated during NREM sleep while higher frequency BLP (*e.g.* theta, gamma) is greater during waking and REM sleep than NREM sleep.

State scoring algorithms consistently rely upon a collection of these electrophysiological features to classify behavioral state (*e.g. Allocca et al., 2019*; *Gao, Turek & Vitaterna, 2016*; *Kreuzer et al., 2015*; *Yan et al., 2017*). For example, by simply using three features (Delta BLP, Gamma BLP and EMG-RMS), we can reliably segregate the majority of epochs from each behavioral state (Fig. 2A). However, as this simple example also shows, many boundary cases remain difficult to classify because a substantial proportion of epochs from each state still overlap within the feature space. To enhance the utility of using electrophysiological features for behavioral state classification, we implemented an artificial neural network (ANN) to predict behavioral state from EEG/EMG features (Fig. 2B). During model training, computational weights are continually altered to produce

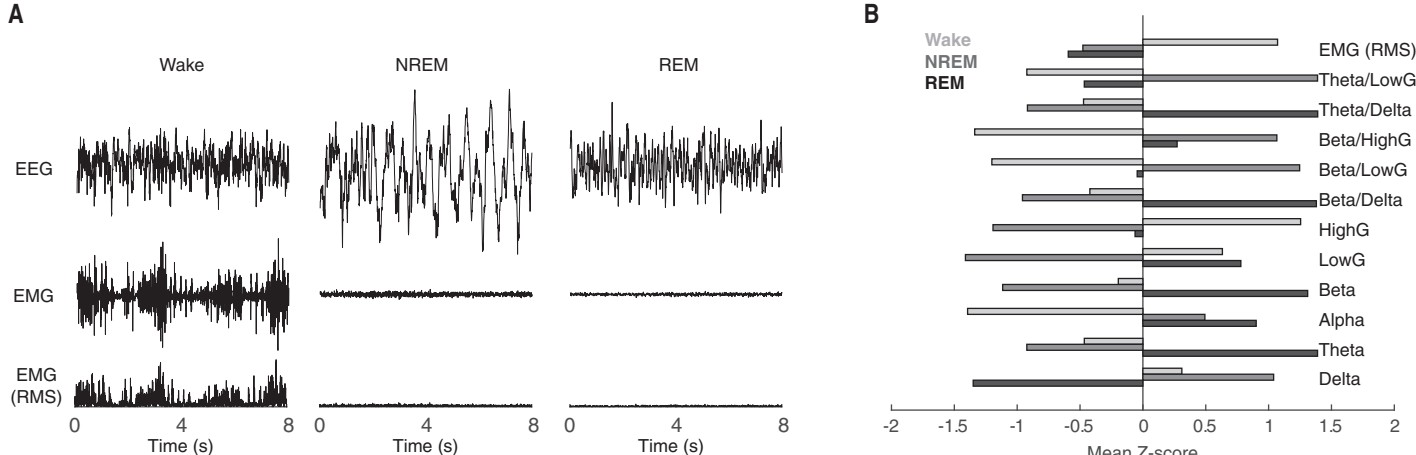

**Figure 1 Characteristic state-dependent differences in electrophysiological activity.** (A) Representative EEG, EMG, and RMS EMG tracings across 8 s periods of wake, NREM sleep, and REM sleep from an individual rat. (B) z-scored electrophysiological features depict state-dependent variation in EEG and EMG activity. For complete description of EEG/EMG feature input to the neural network classifier, see "Methods".

activation maps that accurately predict behavioral state from electrophysiological feature input (see "Methods" and Fig. 2C). Once trained, the ANN can uniquely and nonlinearly combine electrophysiological features to generate distinct activation maps characteristic of each behavioral state. Figure 2D depicts activation maps across all four layers of our ANN in response to feature input from manually scored waking, NREM, and REM epochs. With each progressive layer of the ANN, activation maps converge for epochs derived from the same behavioral state and diverge for those derived from differing states. In doing so, the ANN efficiently and effectively uses electrophysiological features to classify behavioral state without researchers needing to know how to precisely combine feature input, *a priori*.

## An artificial neural network effectively classifies behavioral state

To assess the performance of the ANN detailed above, we compared its behavioral state predictions with manual scoring of the same electrophysiological data by trained undergraduate researchers. Figures 3A–3B depict behavioral state hypnograms for a single day using manual and algorithmic scoring, respectively. Here we observe that each scoring method produces a qualitatively similar classification of behavioral state across the 24 h day. Quantification of the model's performance for this day reveals that the ANN predicted behavioral state with high accuracy, sensitivity, and specificity (Fig. 3C). Across 40 recording days from nine rats (Fig. 3D), the ANN predicted behavioral state with an overall accuracy of 90.87 ± 0.31%, high sensitivity (Wake: 89.82 ± 0.50%, NREM sleep: 92.81 ± 0.58% and REM sleep: 86.28 ± 0.99%), and high specificity (Wake: 95.18 ± 0.34%, NREM sleep: 92.71 ± 0.41% and REM sleep: 97.18 ± 0.19%). Thus, our overall model performance is comparable with that of other high-performing sleep scoring algorithms despite (a) the small amount of training data needed (560, 4 s epochs; 2.6% of day's total) and (b) in contrast to some algorithms, the ANN predicts behavioral state for all test epochs including those manually-scored as artifact (see Table 1). Moreover, the ANN

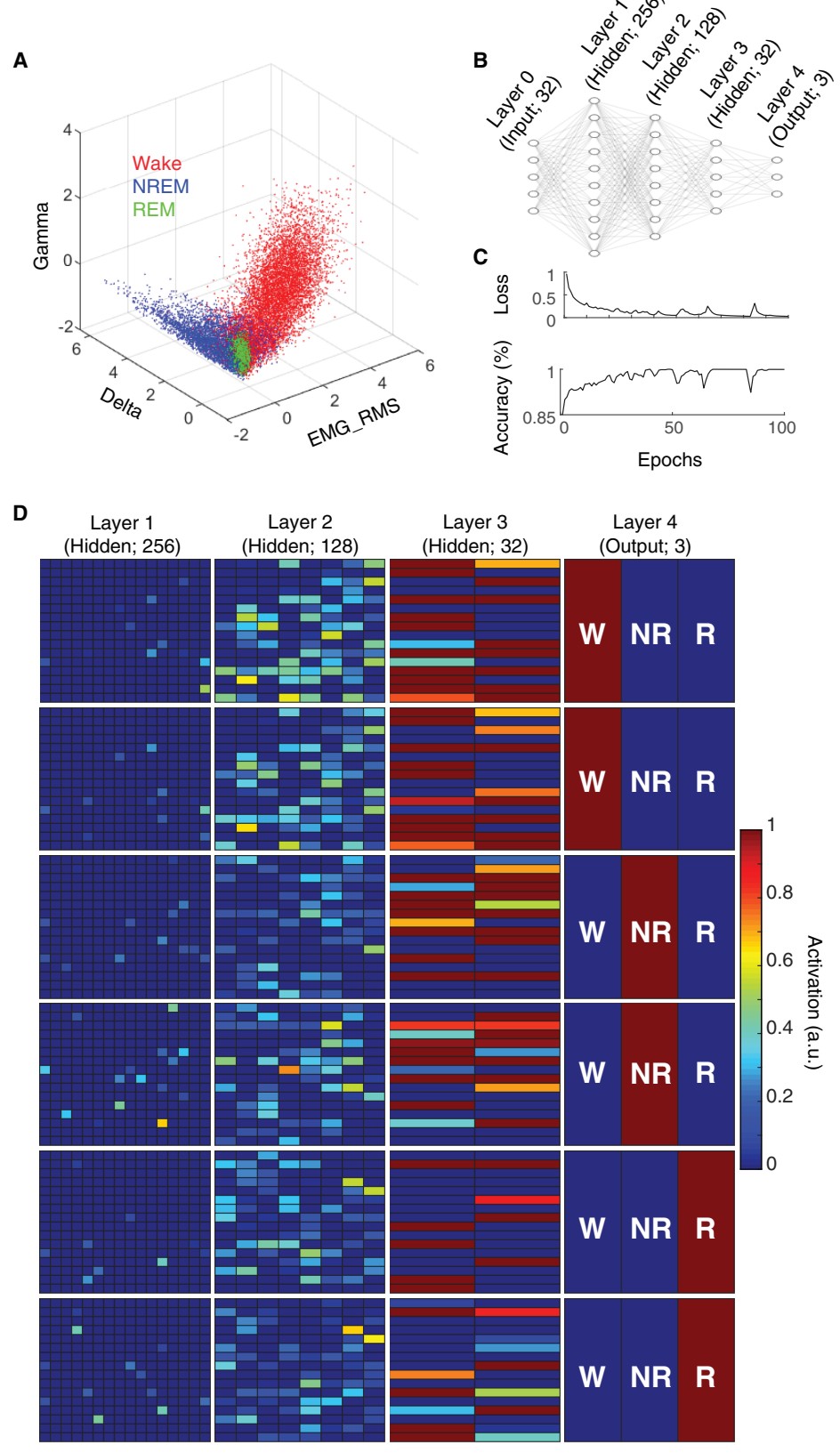
**Figure 2 Artificial neural network reliably predicts behavioral state from EEG and EMG features.** (A) Example data from an individual rat depicts how electrophysiological features alone reliably segregate behavioral state for most 4 s epochs, but fail to resolve ambiguity of many boundary cases. (B) Artificial neural network architecture employed in the current study to enhance the predictive utility of EEG/EMG features for behavioral state classification (see "Methods" for detailed characteristics of input features used). (C) Reductions in categorical cross-entropy loss and increased prediction accuracy of training data occur across model training. Data depict training of a single model. (D) Activation maps across the four layers of our artificial neural network in response to EEG/EMG features. Within each layer, artificial neurons are depicted as colored boxes with color representing the amount of activation of that neuron in response to input features from a 4 s epoch. Each of the six rows depict activation maps from a unique 4 s epoch from the same rat; two epochs of each behavioral state were chosen to show similarities and differences in activation within and across data from each behavioral state. Note, activation of specific neurons in the output layer corresponds directly with the predicted output of the model which matches the manual scoring for each of the epochs presented above.

accomplishes this high performance with rapid computational time (average total run time including fast Fourier transform, feature selection, model training, and prediction output of 223.0 ± 1.1 s when run on a 2.4 GHz Quad-Core Intel Core i5 processor with 8GB 2133 MHz LPDDR3 RAM).

Although the epochs used to train the ANN models characterized above were randomly selected as described in the methods, the overall performance of these models could nevertheless be dependent upon the specific training epochs that happened to be selected. To address this possibility, we repeated the analyses described above five times for each recording, each time using a different random selection of training epochs. Random differences in training epochs selected produced minimal alterations to model performance; between models trained on different random epochs, low average standard errors were observed for overall accuracy (0.25 ± 0.02%), sensitivity (Wake: 0.55 ± 0.05%, NREM sleep: 0.53 ± 0.04% and REM sleep: 1.81 ± 0.16%), and specificity (Wake: 0.51 ± 0.05%, NREM sleep: 0.46 ± 0.04% and REM sleep: 0.24 ± 0.02%). Thus, model performance appears largely unaffected by which specific 2.6% of epochs researchers select for model training.

Although our model compares favorably with other automated scoring approaches, its intended purpose is to serve as a functional tool for sleep researchers. Therefore, more direct assessments of the functional consequences of its use may additionally serve as important metrics of performance. Consequently, across our entire data set, we compared common sleep parameters calculated using our ANN scoring with those calculated using manual scoring. Both scoring methods yield characteristic state-dependent power spectra (Fig. 4A); power spectra exhibit clear 1/f power law scaling (*Bédard, Kröger & Destexhe, 2006*), prominent peaks within theta frequencies during waking and REM sleep, and pronounced slow wave activity (SWA) during NREM sleep. Overall, we observed expected significant differences in power spectra as a function of (1) behavioral state ($F(2,16) = 47.26$, $p = 1.9 \times 10^{-7}$) and (2) frequency ($F(77,616) = 72.18$, $p = 2.32 \times 10^{-260}$), but did not observe a significant effect of scoring method ($F(1,8) = 0.07$, $p = 0.79$). Furthermore, we observed a canonical homeostatic decline in SWA across the light period regardless of scoring method employed (Fig. 4B; effect of light

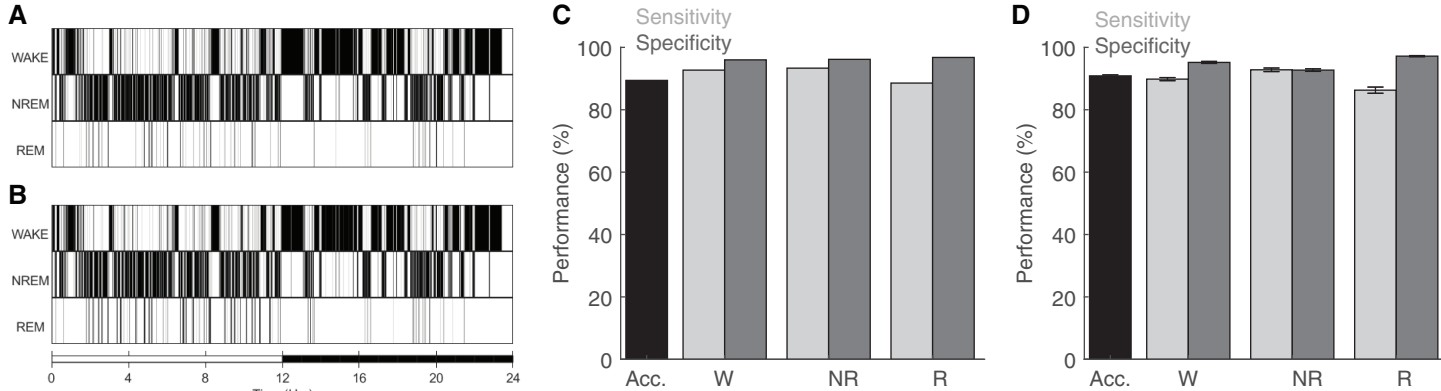

**Figure 3 Sleep scoring performance of artificial neural network.** (A–B) Example hypnograms from an individual rat depict behavioral state as characterized by manual or algorithmic scoring, respectively. Bottom bar depicts the 12/12 h daily light cycle. (C) Quantification of algorithm accuracy, sensitivity, and specificity for example data presented in (A). (D) Accuracy, sensitivity, and specificity of algorithmic scoring across all days ($N = 40$ total from nine rats). Acc—Overall model accuracy; W—Waking; NR—NREM sleep; R—REM sleep.

period time: $F(11,88) = 12.88$, $p = 3.51 \times 10^{-14}$; effect of scoring method: $F(1,8) = 3.63$, $p = 0.09$). Thus, key electrophysiological characteristics of behavioral state and homeostatic sleep are not significantly affected by the use of this ANN for behavioral state classification. Lastly, we examined whether ANN classification altered total duration within each behavioral state (Fig. 4C) and average episode duration (Fig. 4D). During the dark period, the amount of time spent within each behavioral state significantly varied as expected ($F(2,16) = 618.92$, $p = 7.03 \times 10^{-16}$), and this effect was not significantly altered as a function of scoring method ($F(2,16) = 1.97$, $p = 0.17$). In the light period, however, observed significant differences in state duration ($F(2,16) = 481.88$, $p = 5.06 \times 10^{-15}$) were significantly affected by scoring method ($F(2,16) = 15.83$, $p = 1.6 \times 10^{-4}$). Post-hoc $t$-tests reveal that this effect appears largely driven by a significant increase in REM sleep scored by the ANN ($24.32 \pm 4.60\%$ more light period REM) at the expense of a significant decrease in light period waking ($-9.00 \pm 1.58\%$). Mean episode durations across the entire day, however, were not significantly affected by scoring method (Fig. 4D; $F(1,8) = 0.42$, $p = 0.54$). Thus, the use of this ANN for behavioral state classification does not appear to significantly affect most common sleep parameters, with the exception of an increased prevalence of light-period REM duration that arises at the expense of waking duration.

## Optional approaches enhance scoring accuracy or reduce scoring time

Ideally, automated sleep scoring algorithms afford significant time savings with minimal classification error. To this end, we have shown above that our ANN scoring approach classifies behavioral state with high accuracy, sensitivity, and specificity, while only requiring researchers to manually score 2.6% of the day. Indeed, increasing the size of the initial training set beyond 2.6% of the day did not appreciably increase model performance (Fig. S1). Certain research applications, however, may necessitate enhanced accuracy even at the expense of additional scoring time. Alternatively, analyses of very large data sets

**Table 1 ANN model performance as compared to previous automated sleep scoring approaches to classify rodent behavioral state.**

| Source | Approach | Overall accuracy (%) | Manual scoring (% of total epochs) | Freely-available, open source (environment) | Include epochs with visually-scored artifact? |
|---|---|---|---|---|---|
| Current manuscript | Artificial neural network | 91 | 2.6 | Yes (R) | Yes |
| Current manuscript | Artificial neural network | 92 | 2.6 | Yes (R) | No |
| Current manuscript | Artificial neural network + manual rescore | 93 | ~14 | Yes (R) | Yes |
| Current manuscript | Artificial neural network + same rat generalization | 89 | 2.6 (training day) 0 (other test days) | Yes (R) | Yes |
| Exarchos et al., 2020 | Convolution neural network | 93 | ~15 | Yes (Google colab) | Yes |
| Exarchos et al., 2020 | Dimension reduction + clustering | 89 | 0; unsupervised | Yes (Google colab) | Yes |
| Yamabe et al., 2019 | Convolution neural network + long short-term Memory | 97 | Not reported | Yes (Python) | Yes |
| Miladinović et al., 2019 | Convolution neural network + hidden Markov model | 93–99 | ~9 | Yes (torch) | No |
| Miladinović et al., 2019 | Convolution neural network + hidden Markov model | 89 | ~9 | Yes (torch) | Yes |
| Barger et al., 2019 | Convolution neural network + mixture z-scoring | 97 | ~1–2 | Partially[2] (Matlab—GUI) | Yes |
| Allocca et al., 2019 | Support vector machine | 0.94[1] | <1 | Partially[2] (Matlab–GUI) | No |
| Yan et al., 2017 | Threshold decision tree | 91 | 0; Threshold | No (Matlab) | Yes |
| Gao, Turek & Vitaterna, 2016 | Multiple classifier system | ~95 | ~9 | No (Matlab) | No |
| Kreuzer et al., 2015 | Threshold decision tree | 91 | 0; Threshold | Partially[2] (LabVIEW—GUI) | No |
| Bastianini et al., 2014 | Threshold decision tree | 89–97 | 0; Threshold | No (Matlab) | Yes |
| Yaghouby, O'Hara & Sunderam, 2016 | Hidden Markov model | 90 | 0; unsupervised | No (Matlab) | ~5% of epochs excluded |
| Rytkönen, Zitting & Porkka-Heiskanen, 2011 | Naïve Bayes classifier | 93 | 5 | No (Matlab) | Only days with <5% artifact |
| Gross et al., 2009 | Threshold decision tree | 80 | 0; Threshold | Partially[2] (Matlab—GUI) | Yes |
| Stephenson et al., 2009 | Threshold decision tree | 89 | 0; Threshold | Yes (Spreadsheet) | No |
| Crisler et al., 2008 | Support vector machine | 96 | ~4 | No (Matlab) | No visually-scored artifacts |

Notes:
[1] Alternate accuracy metric.
[2] Standalone open-source tools enable algorithm implementation, but not editing.

may benefit from reduced scoring time as long as classification performance is not severely diminished. Below we characterize two optional approaches available with our ANN to satisfy these contrasting needs.

In addition to producing a categorical behavioral state prediction for each epoch, our ANN generates an estimate of the certainty of each prediction. To explore how model certainty relates to model performance, we calculated sensitivity and specificity metrics for each behavioral state as a function of model certainty (Fig. 5A). Model sensitivities (Wake: $r(238) = 0.80$, $p = 1.10 \times 10^{-54}$; NREM: $r(238) = 0.90$, $p = 8.69 \times 10^{-88}$; REM: $r(238) = 0.81$, $p = 4.32 \times 10^{-57}$) and specificities (Wake: $r(238) = 0.75$, $p = 1.33 \times 10^{-44}$; NREM: $r(238) = 0.64$, $p = 4.68 \times 10^{-29}$; REM: $r(238) = 0.83$, $p = 2.75 \times 10^{-62}$) were all significantly

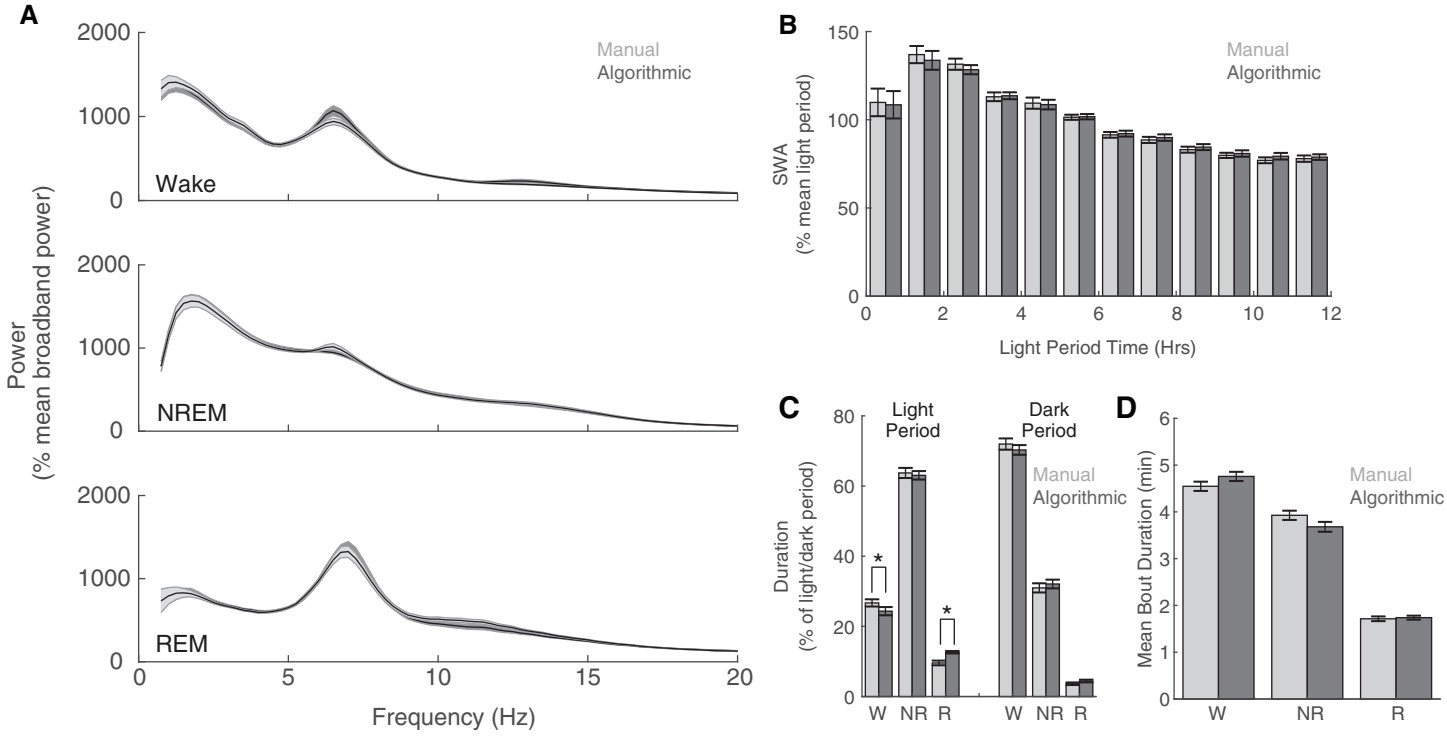

**Figure 4 Quantification of common sleep parameters is highly similar when using manual or algorithmic scoring.** (A) No significant differences in state-dependent power spectra were observed when computed using manual/algorithmic scoring methods. Each line depicts mean +/− standard error. (B) Slow wave activity (SWA) exhibits a similar homeostatic decline across the light period with both scoring methods. (C–D) Total durations and mean bout durations are similar across both scoring methods, with the exception of a significant increase in light-period REM duration and decrease in light-period waking duration. *p < 0.05 W—Waking; NR—NREM sleep; R—REM sleep.

correlated with model certainty. Moreover, these analyses reveal that the vast majority of epochs (88.03 ± 1.23%) were predicted with model certainty greater than 90%. These observations raise the possibility that manually rescoring the relatively small proportion of lower certainty epochs could greatly enhance classification performance. As evident in Fig. 5B, manually rescoring epochs associated with lower prediction certainty provides a focused approach that (1) increases sensitivity and specificity across each behavioral state and (2) still affords a significant reduction in researcher time as the ANN automated scores are used for the vast majority of epochs (see Table 1).

Lastly, although the base implementation of our algorithm only requires researchers to manually score 2.6% of the original file, these small manual scoring requirements may nevertheless present significant time constraints when analyzing very large datasets. Since electrophysiological features associated with each behavioral state are likely to share similar characteristics across recording days and subjects, our trained ANN may be generalizable. To test this possibility, we first trained the ANN on one recording day and then used the trained model to predict behavioral state across all other recording days from the same rat (Fig. 6A). Under these conditions, the ANN model maintained high levels of accuracy, sensitivity, and specificity, albeit with a small drop in performance relative to when tested on the same day as trained. Small, but statistically significant (paired *t*-tests), reductions in overall model accuracy (−1.74 ± 0.41%), NREM sensitivity (−1.95 ± 0.62%)

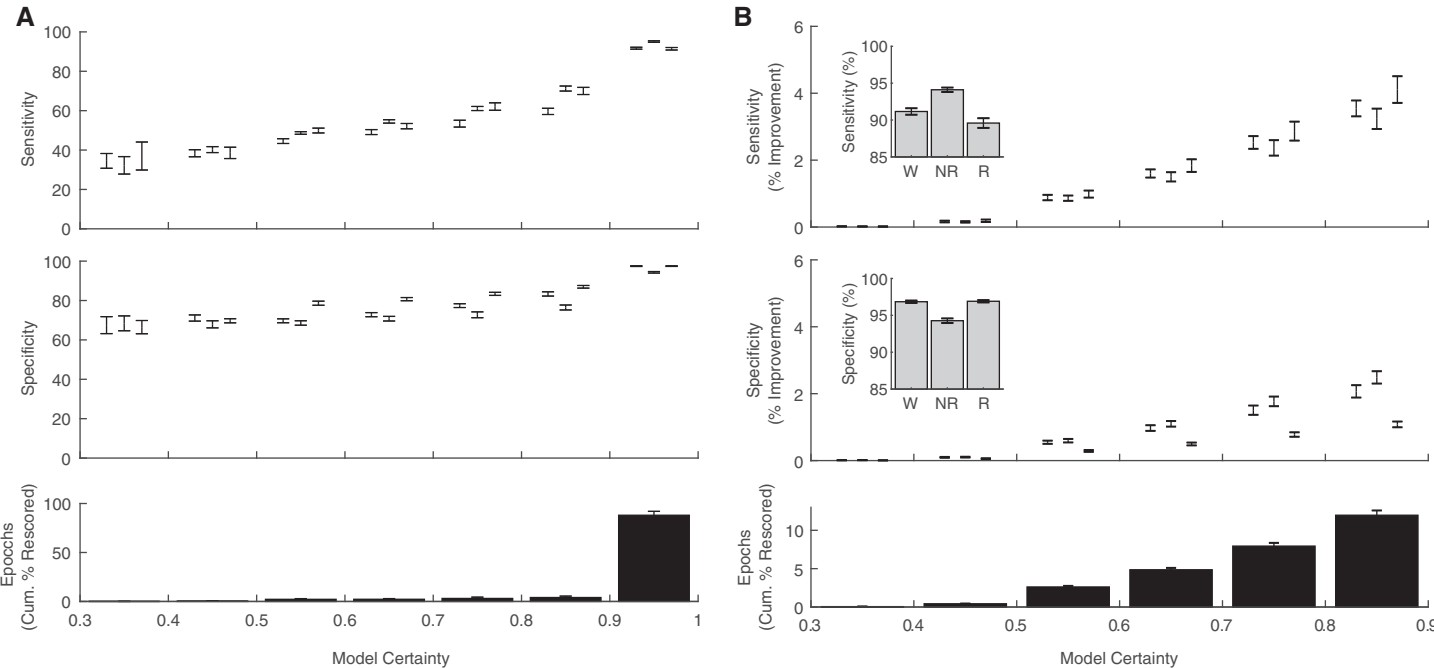

**Figure 5 Algorithm performance corresponds with individual epoch prediction probabilities.** (A) The algorithm has high confidence (>90%) for the majority of the 4 s epochs scored across 24 h and therein produces predictions with high sensitivity and specificity. (B) Sensitivity/Specificity across the entire day are enhanced through manual rescoring of the subset of epochs that have lower (<90%) prediction confidence. Insets depict overall model performance after manual rescoring all epochs with lower prediction confidence. Note sensitivity/specificity in (A) reflect only those epochs within the model certainty bin while in (B) reflect performance across all epochs. Within each bin, data triplets reflect wake, NREM sleep, and REM sleep respectively.

and Waking/NREM specificity (−2.38 ± 0.77% and −1.24 ± 0.57%, respectively) were observed. Waking sensitivity and REM specificity were not significantly affected, while a more pronounced and statistically significant decrease in REM sensitivity was present (−6.42 ± 2.17%). Collectively, these results appear to indicate that the ANN can effectively classify multiple recording days from the same subject when trained using epochs from a single day. We then sought to determine whether an ANN trained on data from one rat could generalize to recordings from other rats (Fig. 6B). While the ANN still maintains moderate performance under these conditions for most parameters of interest (*e.g.* overall accuracy: 85.33 ± 0.40%), significant decreases in performance were observed relative to ANN models trained on the same day as tested in all performance metrics except REM specificity. REM sensitivity was particularly affected with this approach and significantly decreased by −27.50 ± 2.52%. Thus, the ANN appears limited when generalizing to recordings from different rats.

## DISCUSSION

We have developed an artificial neural network that uses EEG/LFP and EMG features to classify behavioral state in rats with high accuracy, sensitivity, and specificity. Quantification of common sleep research parameters (*e.g.* power spectra, SWA, bout duration, and state durations) is largely unaffected when using this ANN scoring as compared to manual scoring. The ANN requires minimal manual classification, has low
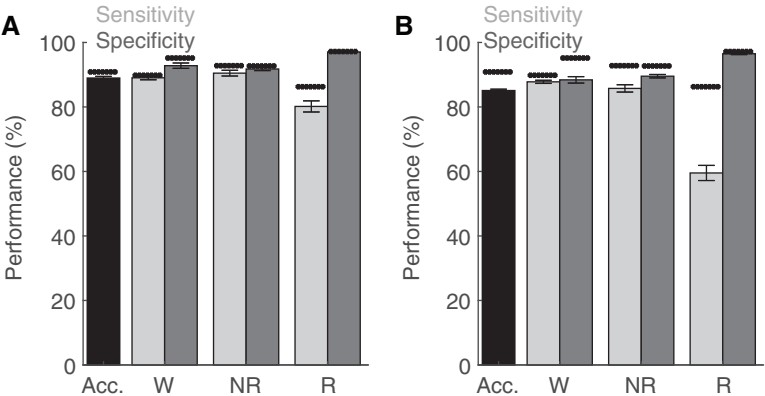

**Figure 6 Classification performance of the trained algorithm when tested across different recording days and rats.** (A–B) State-dependent classification accuracy, sensitivity, and specificity when testing the algorithm on different recording days from the same rat or different rats, respectively. Dark asterisk bars depict mean model performance when tested on the same day as trained (*i.e.* Fig. 3D). Acc—Overall model accuracy; W—Waking; NR—NREM sleep; R—REM sleep.

computational complexity, and critically is readily implementable using freely available software. Optional features enable researchers to prioritize overall accuracy through manual rescoring of epochs identified as low certainty or further reduce the need for manual scoring by generalizing trained networks across multiple recording days of the same rat. Consequently, this ANN appears to serve as a valuable research tool that increases researcher accessibility to effective and efficient algorithmic classification of behavioral state.

Highly accurate behavioral state classification is difficult to achieve despite playing a critical role across a diverse set of research questions. Visual scoring of recorded electrophysiological signals is perhaps the most common approach to solve this challenge yet requires extensive researcher training/time and nevertheless exhibits relatively low inter-rater reliability (83–95%; *Crisler et al., 2008*; *Gross et al., 2009*; *Miladinović et al., 2019*; *Rytkönen, Zitting & Porkka-Heiskanen, 2011*). Recent advances in machine learning have led to the development of algorithmic classifiers that achieve comparable performance to manual scoring yet afford significant reductions in classification time. Of these, deep learning algorithms have achieved the highest accuracies, yet require extensive computational resources and large training sets (*Miladinović et al., 2019*; *Yamabe et al., 2019*). For example, the accuracy of one of these deep learning algorithms is reduced from 96.6% to 80.5% when the training set is reduced from 4,200 files to 500 files (*Yamabe et al., 2019*). These requirements may severely diminish the utility of such approaches for researchers who do not have access to such resources.

By combining supervised learning approaches with deep learning algorithms that utilize simplified neural network architectures, excellent classification performance can be achieved while significantly reducing the need for extensive computational resources as detailed above. Indeed, this approach has previously been successfully implemented for

classification of behavioral state in humans (*Ronzhina et al., 2012*; *Schaltenbrand, Lengelle & Macher, 1993*). Our ANN extends this approach for the classification of behavioral state in rats and achieves high accuracy (90.9%) despite requiring minimal manual scoring (2.6% of daily epochs) and low computational resources. During training, ridge regularization and oversampling were undertaken to overcome the imbalanced distribution of behavioral states and increase classification accuracy (*Janocha & Czarnecki, 2017*). Z-score normalization of feature input helped alleviate misclassification errors that could arise from distributional shift (*Barger et al., 2019*). Post-training application of heuristics to transform a subset of the ANN predictions as a function of previous sleep/wake history was used to further improve classification accuracy. Lastly, through automated identification of individual epochs that were predicted with <90% certainty by the ANN, our algorithm facilitates optional manual rescoring of a subset of epochs that ultimately increases classification accuracy to ~93%.

Although our ANN achieves high overall accuracy, its performance is more limited in terms of REM classification; as compared to manual scoring, the algorithm predicts significantly more REM sleep during the dark period at the expense of waking and only obtains 86.3% sensitivity for REM epochs. Such performance mirrors that of many different automated approaches as REM classification performance is consistently lower than that for other behavioral states (*Exarchos et al., 2020*; Rytkönen et al., 2011; *Yamabe et al., 2019*). Distinguishing between REM sleep and quiet wakefulness presents a significant challenge for automated classifiers because considerable overlap of electrophysiological features is present across these two distinct states. Maximizing feature differences between these states, however, can improve performance. As previously reported (*Allocca et al., 2019*; *Barger et al., 2019*; *Bastianini et al., 2014*), we observed that successful classification was highly dependent upon the EMG signal quality. REM sensitivity of our algorithm was correlated with EMG signal quality (Fig. S2) with REM sensitivity reaching a maximum of 97.8% for an individual day. Additional features, like heart-rate variability, may further differentiate waking and REM sleep and in doing so improve automated classification (*Chouchou & Desseilles, 2014*; *Herzig et al., 2018*). In the absence of clear feature differentiation, our results indicate that manual rescoring of uncertain epochs represents a practical solution for improving performance, achieving 89.59 ± 0.66% average REM sensitivity. Despite some limitations surrounding REM sensitivity, our ANN affords significant benefits as a functional tool for researchers. The quantification of key characteristics inherent to sleep research, including state-dependent power spectra and homeostatic patterns of slow wave activity, are not significantly affected by using this automated scoring approach (Fig. 4).

Unlike most previous automated behavioral state classifiers (*Barger et al., 2019*; *Crisler et al., 2008*; Gao et al., 2016; *Gross et al., 2009*; *Miladinović et al., 2019*; Rytkönen et al., 2011; *Yan et al., 2017*), code for our ANN is not only open source but fully written within a commonly used, freely available software environment (*R Core Team, 2020*). These characteristics enable future researchers to use, and even customize, our initial ANN to meet their specific research needs. Given the similarities in electrophysiological activity

and behavioral state classification in rats and mice, we would expect that the current ANN would perform well when classifying behavioral state in both species. Alternatively, the number of neurons in the ANN output layer could be readily altered to classify additional states characteristic of monkey and/or human sleep (*Hsieh, Robinson & Fuller, 2008*; *Malafeev et al., 2018*), NREM-REM sleep transitions (*Benington, Kodali & Heller, 1994*; *Gross et al., 2009*), and/or pathological states such as cataplexy (*Exarchos et al., 2020*). Additionally, feature input could be amended to include novel features like heart-rate variability (*Chouchou & Desseilles, 2014*; *Herzig et al., 2018*) to enhance classification of states of interest. Lastly, even the base architecture of the presented classifier can be modified; although we altered model hyperparameters (*e.g.* layer number, number of neurons per layer, activation functions, etc.) during model development to enhance classification accuracy, there may be combinations of hyperparameters that we did not test that would yield significant improvement to the model's predictions. Thus, the inherent flexibility of our ANN and ease of access and implementation may enhance the utility of this approach for addressing diverse research questions. Consequently, this ANN represents a valuable tool that can facilitate adoption of highly efficient and accurate automated behavioral state classification.

## CONCLUSIONS

Despite recent advances in automated sleep scoring approaches, significant functional barriers still limit their widespread adoption. Here, we present an efficient and accurate automated sleep scoring algorithm that: (1) has minimal computational needs, (2) is freely available, and (3) is coded within a fully open source environment. The use of this behavioral state classifier did not significantly affect most parameters (*i.e.* state durations, bout lengths, slow wave activity) commonly used in sleep research. Consequently, this accessible and readily implementable ANN may serve as a useful tool for diverse research dependent upon accurate behavioral state classification.

### Funding
The authors received no funding for this work.

### Competing Interests
The authors declare that they have no competing interests.

### Author Contributions
- Jacob G. Ellen conceived and designed the experiments, performed the experiments, analyzed the data, authored or reviewed drafts of the paper, and approved the final draft.
- Michael B. Dash conceived and designed the experiments, performed the experiments, analyzed the data, prepared figures and/or tables, authored or reviewed drafts of the paper, and approved the final draft.

## Animal Ethics

The following information was supplied relating to ethical approvals (*i.e.*, approving body and any reference numbers):

This research was approved by Middlebury College's Institutional Animal Care and Use Committee (Protocol #316-17).

## Data Availability

The raw data is available at Zenodo: Jacob G. Ellen, & Michael B. Dash. (2021). EEG/LFP/EMG data from freely-behaving rats across the sleep/wake cycle (includes manual identification of behavioral state in 4 s epochs) (Data set). Zenodo. DOI 10.5281/zenodo.5227351.

## Supplemental Information

Supplemental information for this article can be found online at http://dx.doi.org/10.7717/peerj.12127#supplemental-information.

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
