# Peer review of "An artificial neural network for automated behavioral state classification in rats"

_PeerJ, doi:10.7717/peerj.12127_

## Round 0.1 · original submission · Major Revisions

Dear Dr. Dash,

We have now received reviews from your article, "An Artificial Neural Network for Automated Classification of Behavioral Status in Rats". The reviewers raised some methodological points to address before making a final decision on publishing your article.

Please find their comments below.

Sincerely,
Emiliano Brunamonti

Reviewer 1 ·

Basic reporting

no comment

Experimental design

This study provides a novel artificial neural network that classifies behavioral states in freely behaving rats. The network receives in input electrophysiological features, previously extracted, that are subsequently processed by two fully connected hidden layers. The output of the network consists of 3 units, each representing the probability that the input data was recorded during the wake, REM, or non-REM state. The performance of the network in classifying the correct state is high and it was assessed after cross-validation. This tool is promising because it is not computational expensive, it is free, and open source. It is a helpful tool to speed up the time necessary to label the behavior state, particularly addressed to sleep researchers.

In my view the following points should be addressed in the revision to improve the manuscript further.

Major points:

1. Even though the description of the architecture of the network is clear in terms of number of hidden layers and number of units per layer, it would be helpful for the reader if the authors could describe why and how they ended up in choosing this particular configuration of the network, i.e. 4 fully connected layers (256, 128, 32, and 3, line 136). Did the authors investigate any other configurations of a feedforward network, i.e. different number of hidden layers and/or different number of units per layer? Is the current architecture the best one in terms of performance and/or time consuming?

2. It would be helpful if the authors could provide an additional figure with the training performance and the loss values as a function of the training epochs (maybe adding another panel into Figure 2 close to the architecture of the network).


3. In the ‘feature extraction’ section (line 147), the authors explain the input to the artificial neural network. According to the architecture of the network, the input consists of 256 units but still, it is not very clear to me what these 256 units are representative of. Different features from EEG-EMG signal are extracted but it is unclear how they sum up to 256 input units. I would suggest spending few more words describing the input structure, and why 256 units as input.


4. The training set was oversampled, as stated in line 167. What is the procedure the authors adopt to oversample the training set?

5. It is not clear what the ‘uncertain label’ defined in line 185 means and how it is further used. A better explanation of the use of the label would be very helpful.

Minor points:

1. Line 163: what does pseudorandom mean in this context? I would say ’50 randomly chosen epochs’

2. Line 205: ‘[…] quality metric is less than 1.67 […]’: why exactly 1.67? It would be helpful if the authors could provide a reason behind the choice of this value for the quality metric threshold.


3. Line 161: ‘[…] initial training set (560, 4s epochs; 2.6% of daily total) […]’. Why did the authors choose exactly 560 4s epochs? It would be helpful to explain in more detail the choice of the size of the training set.


4. Line 227: ‘SWA’ should be defined.

5. When reporting the p-values (for example line 300), if the p-value is significant (much smaller than 0.001), I would suggest writing only the order of magnitude like p=10^-7.


6. Line 333: the following sentence ‘[…] High levels of certainty were associated with excellent model performance’ is not clear because high level is not quantified: is it 70%-80%-90%-100%? I think it is a very general statement that I would personally avoid.

7. Figure 3 C-D: labels of x-axes (i.e., Acc-W-NR-R) should be defined in the caption. Moreover, in figure 3D I would specify the sensitivity-specificity color bar as done in figure 3C.

8. Figure 4B-C-D. Please, provide the legend for the color bars for each figure.

9. Figure 6A-B. Please define the labels of x-axes in the caption.

Validity of the findings

no comment

Reviewer 2 ·

Basic reporting

//

Experimental design

//

Validity of the findings

//

Additional comments

Ms. Ref. No.: PeerJ (#61856)
Title: An artificial neural network for automated behavioral state classification in rats
Jacob Gould Ellen and Michael Bennet Dash

Overview and general recommendation:

The classification of an animal's behavioral state as a function of time is important in the study of neuroscience problems. Usually, this procedure is performed by experimenters in a manual way requiring a lot of time and energy. In this paper the authors propose the use of an artificial neural network to make this procedure semi-automatic (labels must still be manually assigned to a fraction of data for supervised training). In particular the authors test their algorithm considering as input of the network electrophysiology data in freely moving rats and as output the ability of the network to distinguish between awake, REM and NREM states. Although there are already several algorithms in the literature that can solve this task, the authors emphasize that their method is innovative because it is easy to use, open source and computationally inexpensive.

In terms of presentation, I find the article well written and easy to read and understand, the methods are well described so that the reader can reproduce the results themselves and the bibliography is more than comprehensive.

Nevertheless, I found some important (and some trivial) issues that I think are critical to address in order to proceed with an eventual publication in PeerJ that I list below.

Major comments:
1) The network used has 256 input units, but it is not at all clear to me what all these input features are since the authors define only 17 in the text. Which are the others?
2) How does the performance of the network vary when the architecture parameters vary, i.e., number of layers, number of units per layer, different nonlinearities?
3) It is a bit vague how the size of the training set is defined in line 161.
4) Among the features considered as input, which are the most relevant in terms of performance? The purpose of ANNs in general is to find from the learning process the features that are needed for example in a classification problem. Since you already have the relevant features, what is the use of a neural network with different layers? Isn't a perceptron or even a look up table sufficient?
5) I understand that the network has been tested on rat data but the features considered are general. What do you expect from the same ANN architecture on monkey or human data?
6) In line 388 you talk about supervised learning as an alternative approach to deep learning when actually the deep learning in this contest can be seen as supervised learning with number of hidden layers greater than one, then your case falls within the deep learning. In general, it is not clear to me the innovation of this work with the works cited in the literature.
Minor comments:
1) Typo in line 141: output->input.
2) Usually, the term epoch in machine learning is defined as the step in which weights are updated using a batch of data as training, here in this article the term epoch is used to say that different features of electrophysiology are studied on time windows of 4s. I would try to avoid this ambiguity of language replacing epochs with time interval for example.
3) In line 145 it is already mentioned probability, so it is redundant to specify that the sum is one because it must be one by definition of probability.
4) In line 160 it is written "ANN models operate as supervised learning algorithms" as if all the artificial neural networks followed a protocol of supervised learning, this obviously is not true. Change the sentence in "ANN models can operate as supervised learning algorithms".
5) Typo line 183: modal->model.
6) In Figure S1 the caption is missing.
7) In line 227 specify that SWA refers to slow wave activity.

---

## Round 0.2 · accepted · Accept

Dear Dr. Dash,

Both reviewers found your revised version of the document suitable for publication.

Congratulations!
Regards,
Emiliano Brunamonti

Reviewer 1 ·

Basic reporting

no comment

Experimental design

no comment

Validity of the findings

no comment

Additional comments

The authors addressed properly all the comments made by the reviewers. I think the paper is now ready to be published.

Reviewer 2 ·

Basic reporting

Ms. Ref. No.: PeerJ (#61856)
Title: An artificial neural network for automated behavioral state classification in rats
Jacob Gould Ellen and Michael Bennet Dash

Overview and general recommendation:

I find the second version of the article improved and I think the explanations that have been added will be important for anyone who decides to use this algorithm for their research. I am glad that the authors have answered in detail both the major and minor issues I had raised. That said, I am confident that the work is now mature enough to finally be published in PeerJ.

Experimental design

//

Validity of the findings

//

Additional comments

//